# Prevalence of Caries According to the ICDAS II in Children from 6 and 12 Years of Age from Southern Ecuadorian Regions

**DOI:** 10.3390/ijerph19127266

**Published:** 2022-06-14

**Authors:** Eleonor Vélez-León, Alberto Albaladejo, Katherine Cuenca-León, Magaly Jiménez-Romero, Ana Armas-Vega, María Melo

**Affiliations:** 1Department of Surgery, Faculty of Medicine, University of Salamanca, 37007 Salamanca, Spain; albertoalbaladejo@usal.es; 2School of Dentistry, Catholic University of Cuenca, Cuenca 010107, Ecuador; kcuencal@ucacue.edu.ec (K.C.-L.); mjimenezr@ucacue.edu.ec (M.J.-R.); 3School of Dentistry, Hemisferios University, Quito 170527, Ecuador; ana_del_ec@yahoo.es; 4Faculty of Medicine and Dentistry, Department of Stomatology, University of Valencia, 46010 Valencia, Spain; maria.melo.alminana@gmail.com

**Keywords:** child, diagnosis, ICDAS, caries prevalence

## Abstract

In Ecuador, national data on dental caries are scarce and the detection of incipient enamel lesions has been omitted. The objective of this study was to determine the prevalence of caries in school children aged 6 and 12 years of both sexes, belonging to urban and rural areas of three provinces of the country, using the International Caries Detection and Assessment System (ICDAS II). The sample consisted of 665 children from public schools, examined according to ICDASII. Caries prevalence and caries index were established using ICDAS II 2-6/C-G and ICDAS II 4-6/E-G criteria for comparison with WHO indicators. The Mann–Whitney U statistical test was used for comparison of two groups, the effect size was measured with the correlation coefficient. and the Kruskal–Wallis H test (*p* < 0.05) for multiple comparisons. Caries prevalence exceeded 87% for primary and permanent dentition. There were no significant differences according to province (*p* ≤ 0.05). The caries index at 6 years was 6.57 and at 12 years 9.21. The SIC was high at 12 years in rural areas. The prevalence of caries in the population studied was high despite the preventive measures established by health agencies.

## 1. Introduction

The traditional concept of caries as a multifactorial infectious disease has evolved. At present, we know that it is not transmissible and that in addition to the many factors that influence caries development, such as biological, behavioral, psychosocial, and environmental factors, it is also dynamic and mediated by biofilms [1,2].

There are multiple indices to assess caries, whose estimates will depend on several components, such as accuracy, the validity of the measurements, detection criteria, and the definition of the pathology [3,4]. The index chosen will ultimately determine the costs, logistics, and limitations related to the time spent performing the oral examination [5]. These factors could directly affect the results obtained in epidemiological surveys, which are often relevant to the control agencies making decisions about the oral health of each population. The increase in incipient lesions, accompanied by contemporary clinical practice geared toward minimally invasive interventions, has led to the inclusion of initial enamel lesions in the detection system of this pathology through the development of the International Caries Detection and Assessment System (ICDAS) [6,7]. Although the clinical examination time may be slightly longer using ICDAS than using the criteria of the World Health Organization (WHO), this system allows us to estimate the initial caries lesions alongside the planning of preventive strategies and efforts to control disease progression [3,6,8,9,10].

Multiple studies in different countries have made use of this index, leading to clear recommendations on the importance of the diagnosis of incipient lesions [4,5,6,7,11]. This model for the diagnosis and detection of caries is viable, so using it in the public health system would help to reduce the prevalence of caries [7,9].

The few epidemiological studies recorded in Ecuador do indicate DMFT increases with age [12,13]. Studies have been carried out in isolated regions of the country, where it has been determined that in addition to the high prevalence of caries, caries are associated with factors of malnutrition [14,15,16]. In some of the regions of southern Ecuador, such as Azuay, Cañar, and Morona Santiago, the absence of epidemiological studies has not allowed the creation of strategies for the management, control, and follow-up of dental caries. The objective of this study was to determine the prevalence, distribution, and severity of cavitated and non-cavitated lesions in children aged 6 and 12 years of both sexes, belonging to urban and rural zones of Azuay, Cañar, and Morona Santiago regions, using the International Caries Detection and Assessment System (ICDAS II).

## 2. Materials and Methods

This cross-sectional observational study was applied in 2019. The research was approved by the institutional review committee of the Academic Unit of Health and Welfare of the “Universidad Católica de Cuenca” under code No. 048 C.D-2019 (date of approval: 14 February 2019), abiding by the ethical principles of the Declaration of Helsinki and complying with data protection regulations. School principals and parents/caregivers of all children were informed of the objectives and procedure of the study. To authorize the child’s participation in the study, the parent or legal guardian had to sign the informed consent form and submit it before the examination. Children who did not meet the criteria of age, location, and lack of informed consent were excluded from the study.

The dependent, variables calculated for the study were caries prevalence: ICDAS II 2-6/C-G > 0 and ICDAS II 4-6/E-G > 0. Index of decayed, filled primary teeth (dft) ICDAS II C-G and dft/ ICDAS II E-G. Index of decayed, missing, and filled teeth for permanent teeth DMFT-ICDAS II 2-6, DMFT ICDAS II 4-6. The independent variables were sex, age, province (Azuay, Cañar, MoronaSantiago), and environment (urban/rural).

The restoration, morbidity, mortality, and significant caries index (SCI) [17] were calculated from the EG/4-6 codes for both the primary and permanent dentition.

### 2.1. Sample

The sample size was calculated from a population of children equivalent to 183,081 people. This work is part of macro research, which calculated a sample of 1938 participants of children from 6 to 12 stratified proportionally according to the environment by province, with the following slogan 48% students from the urban area and 52% students from the rural area where the children lived. The calculation was made with 99% confidence and 2.5% error. Evaluating the relevance of the sample with an effect size of 0.3, a statistical power of 99.9% was found, with a probability error of 0.043 [18,19].

The final sample for this specific work was 665 public school students residing in three provinces that make up southern Ecuador.

### 2.2. Calibration

The process of calibrating the examiners was directed by professionals certified in the field. The procedure consisted of 3 theoretical sessions, with practical exercises using clinical images and extracted carious teeth, followed by 2 group clinical sessions with 10 schoolchildren of each age group from a local institution. In the clinical part of the calibration process, each examiner reviewed the two groups of children accompanied by a dental student who would assist him/her in recording the information on the forms.

The calibration analysis was performed at the tooth level using the ICDAS II 2-6/C-G, DMFT-dft (ICDAS II 4-6/E-G) classification, with a diagnostic cutoff point at level d1. Agreement between examiners according to Cohen’s kappa was 0.83 for the primary dentition and 0.86 for the permanent dentition.

### 2.3. Examination

Before the examination, the participants’ information was recorded in previously elaborated forms on age, sex, grade, school, ethnicity, location of residence, and type of location (urban, rural).

The test was conducted in classrooms of educational institutions, under standardized conditions recommended by the WHO [20]. In the absence of a dental chair, the children were examined in a chair with their backs straight, and although they were examined in natural light, the examiners used a front light. In each examination, the following were used: a WHO-type periodontal probe, a flat intraoral mirror No. 5, a pair of nitrile gloves, disposable surgical masks for each patient, a headlamp, plus gauze and cotton for moisture control. The calibrated examiners examined while the assistants filled out the data collection form.

### 2.4. Diagnosis of Caries

The caries stage was assessed according to the two-digit ICDAS II criteria [21]. All the dental surfaces of each of the teeth present and measurable were explored: five surfaces for the premolars (permanent dentition) and molars and four for the incisors and canines. Due to the difficulties in obtaining an adequate level of dryness of the dental surfaces during the examination, the epidemiological variant of the ICDAS II criteria was used, which merges codes 1 and 2 [22].

In Ecuador there are no regional studies with the ICDAS II variant, to allow comparison with the WHO criteria, the cutoff point was established at ICDAS II code 4, from this code to code 6, following the WHO criteria.

### 2.5. Statistical Analysis

IBM^®^ SPSS v.27 (New York, NY, USA) and JASP^®^ 0.16.2 (Amsterdam, The Netherlands) statistical programs were used. Descriptive analysis is presented as mean and 95% confidence interval. To establish relationships between variables, nonparametric tests were applied. For comparison between two groups, the Mann–Whitney U test (U) was conducted, the effect size was determined by the biserial correlation coefficient, and for more than two groups, the Kruskal–Wallis H test (H) was performed, as the data were not normally distributed according to the Kolmogorov–Smirnov test (*p* < 0.05). Proportions were compared using the chi-square test (X2) and the effect size was calculated with Cramer’s V statistic (V) and the significance level was 5% (*p* < 0.05), except that for multiple comparisons the Bonferroni adjustment (0.05/3 = 0.017) was applied for a final significance level of *p* < 0.017.

## 3. Results

The final sample for this work was 665 public students from public schools. The 6-year-old group (*n* = 371) was distributed as follows: 135 from Azuay (20.3%), 142 from Cañar (21.3%), and 94 from Morona Santiago (14.1%), while the 12-year-old group (*n* = 294) was distributed as 85 from Azuay (12.8%), 153 from Cañar (23.0%), and 56 from Morona Santiago (8.41%). The participation of both sexes was equal, with 51.2% of males in the 6-year age group and 57.48% in the 12-year age group.

Prevalence and caries rates were calculated using ICDAS II codes 4-6/EG, equivalent to WHO criteria, and ICDAS II codes 2-6/CG. The results for the prevalence of caries are shown in Figure 1.

Figure 1 shows the prevalence of caries, using the ICDAS II 4-6/E-G code in 86% of school-age children and with the ICDAS II 2-6/C-G criterion the values reached 97%.

The dft index (ICDAS II E-G > 0) in 6-year-olds was 6.57, and the DMFT index (ICDAS II 4-6/E-G > 0) was 3.32 in 12-year-olds. When considering all caries codes (ICDAS II 2-6/C-G > 0), the index was 7.08 in 6-year-olds and 9.21 in 12-year-old schoolchildren. Table 1 summarizes the ICDAS II codings according to the most important criteria shared by ICDAS II 2-6/C-G and ICDAS II 4-6/E-G considering all teeth examined (CI = 95%).

In order to compare these results with other studies, following the ICDAS-II E-G/4-6 criteria, the frequency of children with carious (morbidity), filled teeth (restoration), and extracted (mortality) teeth was calculated. In 6-year-old children, the restoration rate was 22.9% (primary dentition), and in 12-year-old it was 55.6%. Of the children aged 6 years (primary dentition), 32.6% had some teeth lost due to caries and 0.7% of the children aged 12 years had some teeth extracted. The rate of decayed teeth was 88.4% in the 6-year-olds and 86.4% in the 12-year-olds. The caries severity index calculated for the 6-year-olds was 7.32 (primary dentition), while for the 12-year-olds it was 8.84.

The analysis of caries prevalence and the different indices with the two criteria in both groups did not represent significant differences according to sex (*p* > 0.05). The distribution of the sample according to environment in the 6-year-old children was 43.9% urban and 56.1% rural, while 45.6% of 12-year-olds lived in urban areas and 54.4% in rural areas. A significant difference (*p* = 0.02) was recorded in the group of 12-year-old students when comparing the DMFT/ ICDAS-II 4-6 index between the urban and rural areas, since adolescents in the rural area had a somewhat higher index (3.72), indicating that the environment had a moderate effect of 0.347 (Table 2). In the group of adolescents, the prevalence of dft ICDAS-II E-G > 1 was much higher in rural areas (23.1%) than in urban areas (14.2%).

By province of residence (Table 3) in the group of 12-year-old adolescents, significant differences were recorded in the dft ICDAS-II/4-6 score. The Bonferroni test revealed that the dft ICDAS-II/E-G differed between the provinces of Cañar and Morona Santiago (*p* < 0.017).

## 4. Discussion

The last epidemiological study carried out in Ecuador in 2009 [23] revealed a high prevalence of carious lesions in the population aged between 6 and 12 years. Since its results were published, many health strategies have been implemented based mainly on the placement of sealants and restorations through the atraumatic restorative technique [24]. Despite these efforts, the results of this study show the high prevalence of carious lesions indicated by the dft and DMFT indices in the assessed population: 3.27 in children 6 years old and 8.11 in children 12 years old.

The rates increased with age, especially in the permanent dentition; in the primary dentition, the presence of carious lesions in severe stages was high, where the components of loss and restoration invite us to think about the need to establish hygiene and education measures for parents or those responsible for the children, so that the prevention strategies that can be established can be effective [25,26]. When considering the presence of caries in the most affected participating population, the caries indices are very high, and the results are very similar to those reported in previous epidemiological studies carried out in various Latin American countries, which show the close relationship between disease symptoms and a lower quality of life of the affected children [27]. These values invite reflection on the measures implemented so far, especially by government entities, when considering the existing scientific evidence demonstrating the influence that fluoride and limiting sugar intake have on dental caries disease [28]. The need for a preventive and follow-up approach is becoming increasingly evident [29].

ICDAS codes as a caries-based detection method have been incorporated in recent years with great success due to their sensitivity and specificity [3], especially when considering carious lesions in subclinical stages [30] and seeking guidelines for treatment [31], which was verified in this study and agrees with previous studies, such as one that showed the socioeconomic situation has a decisive influence on the presence of carious lesions in the child population in Colombia [32]. However, considering that the dft and DMFT indices are the most used in epidemiological studies, along with existing results [33], to minimize errors, we considered ICDAS stages 4–6 the cutoff points, compatible with the WHO criteria [10,32].

The results showed no significant difference in the prevalence of caries between the rural and urban environments overall. However, when considering severity in children aged 12 years, those residing in rural areas showed a much higher rate than children residing in urban areas. The results could be related to the economic and cultural situation [4,34], where the limited access to health services combined with the cost of dental procedures, the low interest of the parents or representatives of the minors, and the absence of knowledge about hygiene and nutritional habits could explain these results [14]. This invites us to reflect as health personnel, to address especially to economically disadvantaged populations, the need to implement educational activities, which seek to fill the gaps in knowledge regarding health, taking advantage of technology, to motivate parents about the oral health of their children [35]. The southern zone of Ecuador has similar economic and cultural characteristics, traditionally affected by an inevitable breakdown of the family component associated with migration [36], which would be disrupting the cycle of knowledge transmission and would explain the similarity between the evaluated provinces.

The absence of epidemiological studies on the national characteristics that have included the ICDAS within the analysis of the presence of caries makes it difficult to control and monitor the results. Previous Ecuadorian studies have been carried out [37], and the results show the need to incorporate this instrument of analysis to train future dentists and project the findings to the strategies to be implemented.

The low presence of restorative components, the high values of missing teeth, and the increasing presence of active carious lesions with age invite us to reflect on changes in the traditionally implemented health strategies, emphasizing preventive strategies targeted at the mother or caregiver as an element of knowledge transmission, prioritizing the practice of tooth brushing accompanied by fluoride-based products [38], and motivating the dentist to take preventive measures from the training stage.

There are no previous studies that include the Ecuadorian population, considering an index as specific and sensitive as the ICDAS, which constitutes one of the strengths of the study; despite this, the study only considered public schools, where the population that attends them generally belongs to an economically underprivileged group, which in a certain way constitutes a bias of the study, which does not allow generalizing the results to the population in general.

## 5. Conclusions

The prevalence of caries is high in the studied population according to ICDAS II criteria. In the primary dentition, there were high caries rates and indexes with low restoration rates. In the permanent dentition, at 12 years of age, caries rates and prevalence were also high both by province and by the environment, although the caries severity index was higher in the urban area.

The use of ICDAS II as a diagnostic tool in studies with large samples implies more examination time and the statistical presentation of the results can lead to different interpretations. However, with the results obtained, we believe that it is important to introduce an early diagnosis of dental caries from the presence of incipient lesions; in this way, it would be possible to reinforce community oral health programs, inserting preventive activities from the first years of life.

## Figures and Tables

**Figure 1 ijerph-19-07266-f001:**
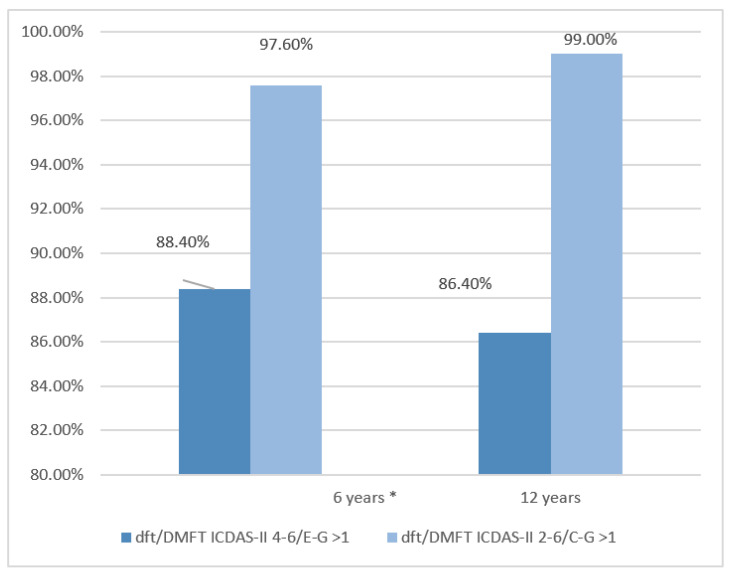
6 years dft/ICDAS-II 4-6/E-G > 1: (IC = 84.9–91.4%) and dft/ICDAS-II 2-6/E-G > 1 (IC = 95.6–98.8%). 12 years DMFT/ICDAS-II 4-6/E-G > 1: (IC = 82.1%; 90.0%) and DMFT ICDAS-II 2-6/E-G > 1 (IC = 97.3–99.7%). * Primary dentition; International Caries Detection and Assessment System (ICDAS); decayed, missing, and filled teeth (DMFT). Confidence interval (CI); DMFT = rate of decayed, missing and filled teeth in permanent teeth; dft = rate of decayed and filled teeth in primary teeth.

**Table 1 ijerph-19-07266-t001:** Indicators according to teeth and age groups.

6 Years *		12 Years
ICDAS 2-C	1.71 (1.56–1.86)	2.35 (2.14–2.56)
ICDAS 3-D	2.29 (2.13–2.45)	4.47 (4.11–4.83)
ICDAS 4-E	1.68 (1.54–1.81)	1.46 (1.33–1.58)
ICDAS 5-F	2.02 (1.85–2.18)	1.7 (1.49–1.91)
ICDAS 6-G	1.87 (1.68–2.06)	2.22 (1.99–2.46)
Permanent decay (ICDAS II 4-6)	0.60 (0.51–0.68)	2.22 (1.94–2.50)
Primary decay (ICDAS II E-G)	3.27 (3.00–3.54)	0.23 (0.17–0.29)
Permanent decay (ICDAS II 2-6)	1.36 (1.22–1.48)	8.11 (7.67–8.55)
Primary decay (ICDAS II C-G)	6.19 (5.86–6.51)	0.24 (0.18–0.30)
Primary fillings	0.36 (0.28–0.44)	0.98 (0.84–1.12)
Primary missing	0.54 (0.44–0.64)	0.01 (−0.01–0.03)
Permanent fillings	0.27 (0.22–0.31)	1.03 (0.89–1.16)
Permanent missing	0.12 (0.08–0.15)	0.07 (0.04–0.10)
dft/DMFT ICDAS-II 2-6/C-G	7.08 (3.79–4.37)	9.21 (8.80–9.62)
dft/DMFT ICDAS-II 4-6/E-G	6.57 (6.24–6.91)	3.32 (3.04–3.59)

* Primary dentition; C, D, E, F, G are the ICDAS II codes for primary dentition.

**Table 2 ijerph-19-07266-t002:** Main caries indicators according to the ICDAS II 4-6/E-G criteria in each age group according to environment.

	6 Years		12 Years	
	Environment		Environment	
	Urban	Rural	Urban	Rural
DMFT/ICDAS-II 4-6 (ICI-ICS)	1.05 (0.87–1.23)	0.92 (0.78–1.07)	2.84 (2.50–3.17)	3.72 (3.30–4.14)
U (*p*)	16484 (*p* = 0.607)		7852 (*p* = 0.002 **)	
Rank-Biserial Correlation	0.028		0.347	
dft ICDAS-II E-G (ICI-ICS)	4.08 (3.64–4.52)	3.08 (3.69–4.46)	1.43 (1.20–1.66)	1.64 (1.43–1.86)
U (*p*)	16484 (*p* = 0.607)		9721 (*p* = 0.044 *)	
Rank-Biserial Correlation	0.043		0.093	
Prevalence of caries in permanent dentition DMFT ICDAS-II 4-6 > 1 (ICI-ICS)	43.6 (36.1–51.2)	42.8 (36.2–49.6)	88.8 (82.7–93.3)	84.4 (78.2–89.4)
χ^2^ (*p*)	0.022 (*p* = 0.882)		1.218 (*p* = 0.270)	
Cramer’s V	0.008		0.064	
Prevalence of caries in primary dentition dft ICDAS-II E-G > 1 (ICI-ICS)	88.3 (82.8–92.6)	88.5 (83.6–92.3)	14.2 (9.1–20.8)	23.1 (17.1–30.1)
χ^2^ (*p*)	0.001 (*p* = 0.972)		3.785 (*p* = 0.052)	
Cramer’s V	0.002		0.113	
General caries prevalence (ICI-ICS)	92.0 (87.1–95.5)	92.8 (88.7–95.7)	90.3 (84.4–94.5)	86.9 (81.0–91.5)
χ^2^ (*p*)	0.076 (*p* = 0.782)		0.836 (*p* = 0.361)	
Cramer’s V	0.014		0.053	

Note: DMFT = rate of decayed, missing and filled teeth in permanent teeth; dft = rate of decayed and filled teeth in primary teeth; ICI = lower confidence interval; ICS = upper confidence interval. * Significant difference (*p* < 0.05); ** significant difference (*p* < 0.01).

**Table 3 ijerph-19-07266-t003:** Main caries indicators under the ICDAS II 4-6/E-G criteria in each age group according to province.

	6 Years			12 Years		
	Azuay	Cañar	Morona Santiago	Azuay	Cañar	Morona Santiago
DMFT/ICDAS-II 4-6 (ICI-ICS)	0.99 (0.80–1.18)	1.08 (0.88–1.28)	0.81 (0.62–1.00)	2.96 (2.52–3.41)	3.82 (3.43–4.22)	2.46 (1.78–3.15)
H	14.123			28.201		
*p*	*p* = 0.058			*p* = 0.034		
dft/ICDAS-II E-G (ICI-ICS)	3.82 (3.40–4.24)	4.47 (3.94–5.01)	3.85 (3.31–4.39)	1.29 (1.00–158)	1.96 (1.76–2.16)	0.80 (0.44–1.17)
H	2.132			33.277		
*p*	*p* = 0.344			*p* = 0.000 *		
Prevalence of caries in permanent dentition DMFT/ICDAS-II 4-6 > 1 (ICI-ICS)	41.5 (33.4–49.9)	49.3 (41.2–57.5)	36.2 (27–46.2)	84.7 (76–91.1)	86.9 (80.9–91.6)	87.5 (77–94.2)
X^2^ (*p*)	4.207 (*p* = 0.122)			0.301 (*p* = 0.860)		
Cramer’s V	0.106			0.032		
Prevalence of caries in primary dentition dft/ICDAS-II E-G > 1 (ICI-ICS)	90.4 (84.5–94.5)	88.7 (82.8–93.1)	85.1 (76.9–91.2)	9.4 (4.6–17)	31.4 (24.4–39)	0.00
X^2^ (*p*)	1.522 (*p* = 0.467)			33.367 (*p* < 0.001)		
Cramer’s V	0.064			0.337 *		
General caries prevalence	94.1 (89.1–97.2)	93.7 (88.8–96.8)	88.3 (80.7–93.6)	88.2 (80.1–93.8)	88.9 (83.2–93.2)	87.5 (77–94.2)
X^2^ (*p*)	3.132 (*p* = 0.209)			0.082 (*p* = 0.960)		
Cramer’s V	0.092			0.017		

Note: DMFT = rate of decayed, missing and filled teeth in permanent teeth; dft = rate of decayed and filled teeth in primary teeth; ICI = Lower confidence interval; ICS = upper confidence interval. * Significant difference (*p* < 0.017) with Bonferroni correction.

## Data Availability

https://ucacueedu-my.sharepoint.com/:u:/g/personal/mvelezl_ucacue_edu_ec/EWCY_ysABVdJl2VgLAcHtMEBqioojuJMGPZoxFaGIaJUAA?e=BCxPGl (accessed on 18 December 2021).

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
