# Peer review of "Prevalence of Caries According to the ICDAS II in Children from 6 and 12 Years of Age from Southern Ecuadorian Regions"

_ijerph, 2022, doi:10.3390/ijerph19127266_

Round 1

Reviewer 1 Report

Generally speaking a clean article with medium-low significance.  I makes its point relative to the title but more information needed in the results and discussion to justify its relativity.  It seems that the aim was to compare the observational data with ICDASII.  In doing so, the research and article make that point which is significant.  

Author Response

Dear reviewer
Thank you for your special contribution to this research work. Please see the attachment.

Reviewer 2 Report

Dear Authors,

Congratulation on your study, let me suggest few changes.

I would recommend to change the position of participants' description from Sample to results .

I think that the date when the approval of the ethics committee was given should be precised.

Author Response

(The authors gave the same response as above.)

Reviewer 3 Report

Thank you for your submission. This manuscript has the potential to add to existing literature; however, it needs some revisions to meet publication quality. Please consider the following points:

-Abstract: the authors should extend the objective to include the relation to gender, environment, and province. More details are necessary concerning the analysis tool and the main tests applied. Also, a conclusion should be added.

-Introduction: the aim of the study is missing.

- Materials and Methods: more details are necessary concerning the used questionnaire and the involved questions, pilot testing?

-Line 71: decayed, filled and decayed teeth needs correction. Also, use primary teeth instead of temporary teeth.

-State who signed the informed consent and whether it was written or not.

-Lines "78-79": "The province of belonging will also be lost as a variable, since each of them has particular cultural characteristics." The sentence should be deleted.

-Sample: The final sample size is unclear, was it 1938 or 665? This part should be clarified. The first paragraph under the subheading "Sample" is confusing. Also, if a sampling formula was used "line 83" include that.

-Calibration: on how many children of each dentition was the calibration done? was it one examiner or more? the sentence in line 99 implies that  there were two or more examiners while line 106 says that it was one.

-Examination: where was the examination done? how did you seat the children for examination?

-Statistical analysis: provide details of the programs.

-Results: this section is dry, incorporate some figures.

-We don't see the results of the Mann-Whitney U test and Kruskal Wallis test in the results (p values) or the tables only those of rank biserial correlation are shown and that was not mentioned in the statistical analysis section.

-Tables: all abbreviations should be described in the footnotes.

- Table 4 is extremely difficult to read. Please display the information in a way that is clear to the reader. Also include all p values.

- The significance level should be revised in the manuscript at times authors mention p<0.05 and at times p<0.01. Check the footnotes of Tables 3 and 4 as this obviously should reflect in the written text.

-Discussion: include the limitations of the study and the points of strength. Also, the study sample was recruited from public schools only (private schools excluded), this factor should be addressed in the discussion.

-Line 176: "These indices increased proportionally with age, in line with previous studies..": not all the indices increased. Check the results of table 2

-Line 199-201: "The results could be related to the economic and cultural situation [4,32], where the limited access to health services combined with the cost of dental procedures, the low interest of the parents or representatives of the minors, and the absence of knowledge about hygiene and nutritional habits could explain these results"

support this sentence with the below recent reference.

"What do parents know about oral health and care for preschool children in the central region of Saudi Arabia?. Pesqui Bras Odontopediatria Clín Integr. 2020; 20:e0103"

- The conclusion is very weak and it does not address the aims or accurately reflect the results. It should be revised after addressing the aim of the study.

Author Response

(The authors gave the same response as above.)

Reviewer 4 Report

The manuscript is well-written and it highlights the prevalence of caries in Sothern Ecuadorian Regions. 

The findings of study are likely to assist the policy makers and healthcare providers while designing the strategies with regard to the prevention and management of dental caries in these particular regions. 

In line 48, the text "click or tap here to enter text'' should be deleted. 

In table 3, the values are not typed properly, therefore the authors are suggested to place the values in an appropriate way. 

Author Response

(The authors gave the same response as above.)

Round 2

Reviewer 3 Report

The manuscript looks better, indeed. These last points are left:

1-Sample: first paragraph about sample size needs a reference number. I suggest to add the sentence in lines 151-152 “Evaluating the relevance of the sample with an effect size of 0.3, a statistical power of 99.9% was found, with a probability error of 0.043” to it and perhaps adding a reference number to it.

2- Improve the visual appearance of Figure 1.

3-Line 165: “Figure 1 shows the prevalence of caries, using the ICDAS II 4-6/E-G code in 86% of school children and with the ICDAS II 2-6/C-G criterion the values reached 97%”. It is better to edit this sentence for clarity “in 86% of school children”.

4-Table 4: make sure that all asterisk symbols are next to the correct values.
